# An Algorithm for Optimizing the Process Parameters of the Spindle Process of Universal CNC Machine Tools Based on the Most Probable Explanation of Bayesian Networks

Liyue Zhang [1], Haoran Liu [1,*], Niantai Wang [1], Yuhua Qin [1] and Enping Chen [2]

[1] The Key Laboratory for Special Fiber and Fiber Sensor of Hebei Province, School of Information Science and Engineering, Yanshan University, Qinhuangdao 066004, China; liyuezhang_94@163.com (L.Z.); niant_wang@163.com (N.W.); yuhua_q@163.com (Y.Q.)

[2] School of Mechanical Engineering, Yanshan University, Qinhuangdao 066004, China; ep_edu@163.com

[*] Correspondence: haoran044@163.com

**Abstract:** As an essential component of a universal CNC machine tool, the spindle plays a critical role in determining the accuracy of machining parts. The three cutting process parameters (cutting speed, feed speed, and cutting depth) are the most important optimization input parameters for studying process optimization. Better processing quality is often achieved through their optimization. Therefore, it is necessary to study the three cutting process parameters of the CNC machine tool spindle. In this paper, we proposed an improved algorithm incorporated with the beetle antennae search algorithm for the most probable explanation in Bayesian networks to achieve optimization calculation of process parameters. This work focuses on building adaptive dynamic step parameters to improve detection behavior. The chaotic strategy is discretized and used to establish the dominant initial population during the population initialization. This article uses four standard network data sets to compare the time and fitness values based on the improved algorithm. The experimental results show that the proposed algorithm is superior in time and accuracy compared to similar algorithms. At the same time, an optimization example for the actual machining of a universal CNC machine tool spindle was provided. Through the optimization of this algorithm, the true machining quality was improved.

**Keywords:** CNC; optimize process parameters; most probable explanation; Bayesian networks; beetle antennae search; adaptive dynamic step parameter





## 1. Introduction

With the arrival of a new industrial revolution, global industrial powers are deploying industrial strategies to accelerate their industrial development, such as the United States' Advanced Manufacturing Partnership Program, Germany's Industry 4.0, Japan's Manufacturing White Paper, and "Made in China 2025" [1,2]. With the continuous development of artificial intelligence technology and industrial big data in the manufacturing industry, especially in CNC machine tool processing, some scholars have used artificial intelligence technology and big data to study the optimization of process parameters and processing quality and have made this research continue [3].

The Computer Numerical Control (CNC) machine tool [4] is one of the most widely used machine tools in machining. CNC machine tools process workpieces such as metal or wood. This process involves manufacturing products by cutting or milling workpieces according to pre-designed shapes. Manufacturing products using CNC machine tools can affect the cutting tools of the CNC machine tool, as the machine tool processes the workpiece while rotating the device fixed on the spindle motor of the CNC machine tool [5,6].

The universal machine tool is one of the most widely used and mass-produced machine tools. Its layout is reasonable and compact, with a small footprint, an independent spindle structure, good rigidity, and vibration resistance. It has a large processing capacity, stable accuracy, and high efficiency. The machine tool control system has strict protective measures to ensure personal and equipment safety. It is also suitable for large cutting capacity, high efficiency, and large-scale processing of parts in mechanical industries such as automobiles, ships, bearings, and military sectors [7].

At present, the process of establishing process parameter optimization models adopts multi-objective optimization algorithms, most of which are based on mechanism models. For example, the multi-objective parameter optimization model for multi-channel CNC and production cost was established based on the energy efficiency mechanism of the milling process [8]. The multi-objective optimization algorithm was studied based on teaching and learning (ITLBO) to optimize cutting and feed speed process parameters [9]. The multi-objective process parameter optimization model was established based on the transient-steady-state energy consumption mechanism with high-quality, low-energy consumption processing of CNC machine tools as the optimization goal [10]. The multi-objective cutting parameter optimization model for multi-pass turning of CNC machine tools was studied based on the cutting energy consumption mechanism to determine the optimal cutting parameters [11]. However, the utilization rate of processing quality data and process parameter data is low, and the relationship between data mining process parameters and quality still needs to be fully utilized. Therefore, this proposed algorithm fully uses data and the most probable explanation to establish an optimization algorithm for process parameters, improving the efficiency of utilizing machine tool manufacturing data.

Bayesian networks (BNs) [12] concisely model probability distributions over a set of random variables. They are self-explanatory, easy to understand, and well-suited for representing causal relationships [13]. BNs have been well recognized as a rigorous methodology for the quantification of risks, uncertainty modeling, and decision-making in the presence of structural dynamics [14]. A key computational problem in BNs is the computation of the most probable explanation (MPE). When the values of observed variables (evidence nodes) are given, MPE is obtained by calculating the values of all non-observed variables (non-evidence nodes), which satisfies the greatest joint probability [15,16]. In Bayesian networks, an explanation of evidence refers to a state combination of all variables in the network consistent with evidence, which is often the explanation with the greatest relationship probability.

The solution of BNs is used in some fields, and examples are as follows: the Bayesian network and neural network were used to model the thermal error of the CNC machine tool feed drive system and study the relationship between the temperature rise and positioning error of the feed drive system [17]. The Bayesian network reliability assessment method was proposed, which considers dynamics and fuzziness. It analyzes the dynamic fuzzy reliability of the balance circuit of the hydraulic system of CNC machine tools and verifies the application of this method in system reliability assessment, providing a basis for CNC machine tool hydraulic system balance circuits. It provides support for fault diagnosis of machine tools [18]. The machined surface quality detection system was developed using CNN architecture and Bayesian optimization, achieving low cost and reliability [19]. The Bayesian network was used to analyze the reliability of related failure modes of CNC machine tools, achieving reliability modeling of complex systems and effective analysis of related failures [20].

Searching MPEs from BNs is an NP-hard question [21,22]. Currently, there are two main methods for solving the problem: the query-based method and the local search-based method. The query-based method is more efficient in small networks, but finding effective solutions in large, complex networks takes a lot of work. The local search-based method can find approximate or optimal solutions in large networks. Still, many of these algorithms easily fall into the local optimum, and the optimization efficiency needs to be improved [23].

This paper proposes a most probable explanation algorithm based on the beetle antennae search algorithm (BAS-MPE). The population algorithm is applied to BN inference, which establishes the framework of MPE. The method of generating the dominant initial population is proposed. At the same time, the adaptive dynamic parameters are used to improve the beetles' detection behavior. Through population iteration, MPE is obtained. In this paper, the algorithm's convergence is proved by the probability measure method. Compared with several similar algorithms, the experimental results show that the proposed algorithm has higher optimization efficiency and better results.

The objectives of this article are to study the most probable explanation algorithm in Bayesian networks and apply it to the optimization of cutting process parameters (mainly including cutting speed, feed speed, and cutting depth) for universal machine tool spindles production to improve the optimization goal (machining quality). This paper provides a reference for the optimization of process parameters in production.

The main contributions of this article are as follows: (1) this study attempts to propose a most probable explanation algorithm in Bayesian networks based on the beetle antennae search for optimizing the spindle process parameters of universal CNC machine tools. Additionally, it was confirmed through experiments that using the spindle process parameters of general CNC machine tools optimized by the proposed algorithm can improve the actual machining quality. (2) The proposed algorithm combines the state information of nodes and uses a chaotic strategy to generate a dominant initial population to increase population diversity. (3) The proposed algorithm uses dynamic step size adjustment parameters in search behavior and detection behavior to balance global search and local development, enhancing the likelihood of the algorithm jumping out of the local optimum.

The organizational arrangement of the remaining parts is as follows: Section 2 introduces the related works on optimization of process parameters for CNC and the most probable explanation algorithm. Section 3 presents the implementation steps of the improved algorithm. In Section 4, the convergence proof of the algorithm is provided. In Section 5, we conducted experiments to verify the algorithm's effectiveness and provide an optimization example of process parameters for the spindle of universal CNC machine tools. Finally, the conclusion is described in Section 6.

## 2. Related Works

### 2.1. Optimization of Process Parameters for CNC

The reasonable selection of process parameters will affect the machining quality during CNC. However, as manufacturing methods continue to evolve and new technologies are continuously introduced, it is crucial to consider the future of CNC machining operations [24]. Process parameters mainly include cutting speed, feed speed, cutting depth, cutting fluid, cutting tools, etc. The three cutting process parameters (cutting speed, feed speed, and cutting depth) are the most critical input parameters for studying process optimization, and it is often possible to achieve better optimization goals by optimizing them [25,26]. In the following, some researchers have used multi-objective optimization and some other methods to study the process parameter optimization problem of CNC machine tools.

The multi-objective particle swarm optimization algorithm was used to establish a multi-objective parameter optimization model for the multi-channel CNC milling process's energy efficiency and production cost to determine the optimal cutting speed, feed rate, and cutting depth and prove that the optimized values are within the acceptable range of the machine [8]. The optimized multi-objective optimization algorithm based on teaching and learning (ITLBO) was studied to optimize the process parameters of CNC cutting, mainly cutting speed and feed speed [9]. The method was proposed based on population optimization by combining the extreme learning machine and the particle swarm optimization algorithm and applying it to the optimization of CNC turning process parameters to achieve control of the cutting speed, feed speed, cutting depth, and surface roughness [25]. The genetic regression neural network GA-GRNN was studied

to design a milling performance prediction model. It optimized the process parameters in the milling process based on this model: spindle speed, number of grinding grooves, and grinding tooth feed [27]. The transient-steady-state energy consumption model was established for CNC lathes, and based on this, a multi-objective model was established with spindle speed, feed rate, and cutting depth as optimization variables, and high-quality, low-energy consumption processing of CNC machine tools as optimization goals [10]. The advanced Tabu search and scatter search were used to establish a multi-objective cutting parameter optimization model for multi-pass turning of CNC machine tools to determine the optimal cutting parameters, including spindle speed, feed speed, cutting depth, and roughness [11]. The OFAT technology was used to conduct CNC milling experiments and determined parameters such as optimal feed speed, cutting speed, cutting depth, and tool tip radius through experiments to ensure the integrity of the cutting surface and good cutting efficiency [26]. The Neural Network (NN) and Multi-objective Swarm Algorithm (MSA) were used to establish a CNC milling processing optimization model. It optimizes the helix angle, axial cutting depth, radial cutting depth, and cutting speed to achieve a better surface finish and minimum tool wear rate [28].

Currently, most of the optimization research on the process parameters of these CNC machine tools can better solve some process optimization problems. This provides a research basis for optimizing and applying CNC machine tool process parameters and better guides the production of enterprises. However, there are also some problems, such as most multi-objective optimization algorithms having optimization weight imbalance and easily falling into local optimum. Some model research focuses on mechanism modeling, and the utilization rate of processing quality data and process parameter data is low. The relationship between data mining process parameters and quality must still be fully utilized.

Therefore, this article fully uses quality data and process parameter data and combines the most probable explanation algorithm to optimize the cutting speed, feed speed, and cutting depth of CNC cutting process parameters to improve the processing quality of CNC.

### 2.2. The Most Probable Explanation of Bayesian Networks

Due to the limitations of the query-based method, experts have mainly used the local search-based method to solve MPE in recent years. In [29], the Ant Colony Optimization algorithm for the most probable explanation (ANT-MPE) is proposed. By constructing the pheromone tables, the heuristic function tables, and the ant decision tables, the effective optimization of the ant colony for MPE is realized. However, the accuracy of the solution depends on the choice of the weight of pheromone trails and the weight of the local heuristic function. Sriwachirawat et al. [30] proposed a niching genetic algorithm (NGA) for MPE, which uses the multi-fractal and clustering characteristics of Bayesian networks to increase the range of population solutions in the early stage of the algorithm. The algorithm makes use of the observation that there are regions within the joint probability distribution of the Bayesian networks that are highly self-similar. In [31], the Stochastic Local Search Algorithm (SLS) to solve MPE is proposed. The algorithm can quickly reach the vicinity of MPE through a random local search strategy, but it is easy for the algorithm to fall into the local optimal value in the later stage. Pillai et al. [32] proposed the probabilistic driving algorithm based on swarm intelligence (DMVPSO) to solve MPE. The algorithm can solve MPE using the particle swarm optimization algorithm when there are few network nodes. However, when the number of network nodes is large, the algorithm is not effective. In [23], the algorithm performs a single iteration of the traditional PSO algorithm for each group. Overlapping clusters compete on their individual state assignments and use communication mechanisms to share information between the clusters. The algorithm can obtain better results than the single population through population competition, but it needs more group competition operation, and its efficiency needs to be improved.

The abovementioned methods are of significant reference value for searching for MPE of BNs. However, these methods have their shortcomings, as mentioned above. For the MPE of BNs, scholars widely use the existing optimization algorithms to solve the

problem directly but rarely improve the applicability and adaptability of the strategy for this problem. It is an excellent idea to transform the MPE problem into the optimal solution problem. The beetle antennae search algorithm (BAS) is a perfect way to solve the optimal situation. At present, BAS is rarely applied in the MPE of BNs. The population optimization algorithm is one of the most effective methods to solve MPE based on the local search method [23,29,31,32].

In this paper, the search strategy in the improved optimization algorithm is used to avoid the algorithm falling into local optimization. The beetle antennae search algorithm [33] is a mathematical model algorithm based on the behavior process of a beetle searching for food. The beetle searches for food sources through search behavior and detection behavior which returns it to the current optimal food source location. Compared with other population algorithms, such as particle swarm optimization algorithm, ant colony optimization algorithm, and bee colony optimization algorithm, the beetle antennae search algorithm inherits the excellent information from the previous generation of individuals effectively, and it avoids the blind randomness of the search. The algorithm does not need more parameters, gradient information, and specific function forms, making the optimization strategy of the algorithm simpler and more suitable for single-objective function optimization. However, the searching phase of the beetle antenna search algorithm only provides two directions, and the optimization effect is poor. The step size of the algorithm is a linear decreasing function, which makes the algorithm fall into the local optimum in the later stage. Therefore, combining the characteristics of the beetle antenna search algorithm, this paper designs a beetle antenna search algorithm, which is the most probable explanation in Bayesian networks. The significant differences between our work and the related works are summarized in Table 1.

**Table 1.** Comparison between the related works.

| Ref. | Nonrandom Initial Population | Improved Optimization Strategy | Population Competition Mechanism |
|:---:|:---:|:---:|:---:|
| [23] | No | Yes | Yes |
| [29] | No | No | Yes |
| [30] | No | Yes | No |
| [31] | Yes | No | No |
| [32] | No | Yes | Yes |
| This work | Yes | Yes | Yes |

Therefore, combining the problems of MPE algorithms and the characteristics of the beetle antenna search algorithm, this paper designs a beetle antenna search algorithm for the most probable explanation in Bayesian networks.

## 3. The Proposed Algorithm (BAS-MPE)

The proposed algorithm BAS-MPE establishes the dominant initial population by introducing a chaotic strategy and maps the strategy to the discrete domain. BAS-MPE improves the search behavior of the beetle and uses the adaptive dynamic parameters to adjust the moving step of the beetle in the detection behavior. Through population iteration, BAS-MPE finds the individual with the highest fitness.

### 3.1. Bayesian Networks and MPE Problem

A Bayesian network (Figure 1) is a Directed Acyclic Graph (DAG) where nodes represent random variables and edges represent conditional dependencies between random variables. Attached to each node is a Conditional Probability Table (CPT) that describes the conditional probability distribution of that node given its parents' states. Distributions in a BN can be discrete or continuous. In this paper, we only consider discrete ones. BNs represent joint probability distributions in a compact manner. Let $B = \{Q_1, Q_2, \cdots, Q_n\}$ be

the random variables in a network. Every entry in the joint distribution $P(Q_1, Q_2, \cdots, Q_n)$ can be calculated using the following chain rule:

$$P(Q_1, Q_2, \cdots, Q_n) = \prod_{i=1}^{n} P(Q_i | \pi(Q_i)), \tag{1}$$

where $\pi(Q_i)$ denotes the parent nodes of $Q_i$. Figure 1 shows a simple BN with four nodes, the Sprinkler network [34].

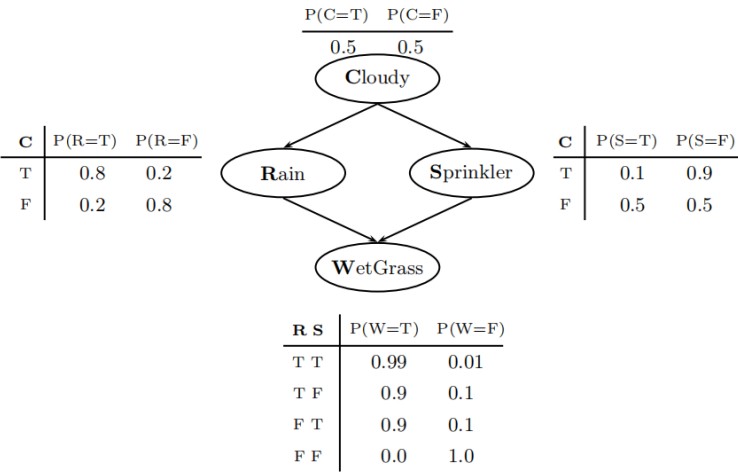

**Figure 1.** The Sprinkler Network.

Let $(B, P)$ be a Bayesian network where $B$ is a DAG and $P$ is a set of CPTs, one for each node in $B$. An evidence $E$ is a set of instantiated nodes. An explanation is a complete assignment of all node values consistent with $E$. Each explanation's probability can be computed in linear time using Equation (1). For example, in the Sprinkler network (Figure 1), suppose we have observed that the grass is wet, i.e., the $E = (W = \text{T})$. One possible explanation of this is: $\{C = T, R = T, S = F, W = T\}$. Its probability is:

$$
\begin{aligned}
& P(C = T, R = T, S = F, W = T) \\
& = P(C = T) \times P(R = T | C = T) \\
& \times P(S = \text{F} | C = T) \times P(W = \text{T} | R = T, S = \text{F}) \\
& = 0.5 \times 0.8 \times 0.9 \times 0.9 \\
& = 0.324.
\end{aligned}
\tag{2}
$$

MPE is an explanation with the highest probability. Given the observed evidence, it provides the most probable state of the world. The algorithm proposed in this paper is to solve the Most Probable Explanation (MPE) problem in Bayesian networks. This network example (the Sprinkler network) is used as a network example. MPE has many applications in diagnosis, abduction, and explanation.

### 3.2. Variable Description and Initial Population

The correspondence between the variables of the population algorithm and Bayesian networks is as follows: the beetle individual $M$: a combination state of all nodes in Bayesian networks (a probable explanation of Bayesian networks), as shown in Equation (3); the beetle population $G$: a cluster representing individual beetles (multiple probable explanations of Bayesian networks), as shown in Equation (4).

$$M = \{X_1 = x_1, X_2 = x_2, \cdots, X_n = x_n\}, \tag{3}$$

$$G = \{M_1, M_2, \cdots, M_m\}, \tag{4}$$

where $X_i, (i = 1, 2, \cdots, n)$ is the $i$-th node of Bayesian networks, $x_i, (i = 1, 2, \cdots, n)$ is the state value of the node $X_i$.

BAS-MPE produces $h$ different beetle individuals $\{M_k(0), (k = 1, 2, \cdots, h)\}$ consistent with DAG. According to Equation (5), BAS-MPE screens to obtain the individuals that, with the evidence nodes, state $\{e_1, e_2, \cdots, e_v\}$ in $E$ from $\{M_k(0), (k = 1, 2, \cdots, h)\}$. We generate non-repetitive individuals of longicorn beetles through the above operation, and combine them into $G(0)$.

$$G(0) = \left\{ M_j(0) = M_k(0), \text{if } (X_1^k(0) = e_1) \cup \cdots \cup (X_v^k(0) = e_v) \right\}, \tag{5}$$

where $X_w^k(0)$ is the $w$-th evidence node of the $k$-th individual, $M_k(0)$ is the $k$-th individual, $w = 1, 2, \cdots, v$.

The chaotic number $z_i \in (0, 1)$ is generated at each node $\left\{ X_i^j(0), i = 1, 2, \cdots, n \right\}$ of $M_j(0) \in G(0)$. All nodes' chaotic numbers form an n-dimensional chaotic initial vector $Z^1 = (z_1^1, z_2^1, \cdots, z_n^1)$ of $M_j(0)$. We generate a discrete chaotic sequence $Z^m = (z_1^m, z_2^m, \cdots, z_n^m)$ by $m - 1$ iterations according to

$$Z^{l+1} = 4Z^l \left( 1 - Z^l \right), l = 1, \cdots, m - 1. \tag{6}$$

BAS-MPE loads the chaotic individuals $M_{j,\text{ch}}(0)$ according to the state range $[a_1, a_2]$ of each node in Equation (7). The algorithm only operates on non-evidence nodes, and the value state of evidence nodes remains unchanged. All the individuals complete the above steps to generate the dominant initial population $G(0)$, as shown in Equation (8). $G(0)$ is used as the first generation of the algorithm.

$$\begin{cases} X_i^j(0) = a_1 + randn(z_i^m(a_2 - a_1)), \\ M_{j,\text{ch}}(0) = \left\{ X_i^j(0) \right\}, \end{cases} \tag{7}$$

$$G(1) = \{ M_j(0), j = 1, 2, \cdots, m \}, \tag{8}$$

where $randn(*)$ means performing rounding on $*$.

*3.3. Search Behavior*

The search behavior of the beetle antennae search algorithm searches the left and right directions. We select different node state values at each node and assign the node state to the left and right antennae separately.

Assuming that there are $m_i$ states of the $i$-th node, each node state of the part (including $v_1$ and $v_2$) between the $v_1$-th node and the $v_2$-th node of the individual $M_j(t)$ is randomly selected. The state of the individual evidence node and other nodes remains unchanged. The left antennae coordinate $M_{j,lt}(t)$ of the individual $M_j(t)$ is obtained in Equation (9).

$$M_{j,lt} = \left\{ X_i = \begin{cases} x_p, \text{if } p \in (v_1, v_2) \cap x_p \notin \{e_1, e_2, \cdots, e_v\}, \\ x_i, \text{else}, \end{cases} \right\}, \tag{9}$$

where $x_i$ is the original state of $X_i$, $x_p$ is the state of node $X_i$ that excludes the original state in any other state.

Execute the above process again. Defining $x_p$ as the state node $X_i$ that excludes the original state $x_i$ and the left antennae coordinate $x_{i,le}$ in any other state. The right antennae coordinate $M_{j,rt}(t)$ for the individual $M_j(t)$ is obtained.

The purpose of solving MPE is to find the node states when all nodes with the highest joint probability are in evidence. So, we use the joint probability that the value is 0 to 1 [22] as the evaluation of fitness. In order to observe the difference in probability more clearly, the logarithmic joint probability is used as the final evaluation of fitness. The fitness function evaluates the joint probability of all nodes' state distribution. We calculate the

fitness of $M_{j,lt}(t)$ and $M_{j,rt}(t)$ by Equation (10) (the fitness of $M_{j,lt}(t)$ is $f_{j,lt}(t)$ and the $M_{j,rt}(t)$ is $f_{j,rt}(t)$).

$$f = lg\left(\prod_n^{i=1} P(X_i|Pa(X_i))\right), \tag{10}$$

where $Pa(X_i)$ is the parent node of $X_i$.

Substitute $f_{j,lt}(t)$ and $f_{j,rt}(t)$ into Equation (9). It selects the larger fitness of the left and right coordinates as the better coordinate $M_{j,better}(t)$ in the $t$-th generation.

$$M_{j,better}(t) = \begin{cases} M_{j,lt}(t), & \text{if } f_{j,lt}(t) > f_{j,rt}(t), \\ M_{j,rt}(t), & \text{else.} \end{cases} \tag{11}$$

Two crossover nodes ($X_{c_1}^j(t)$ and $X_{c_1}^j(t)$, ($1 \leq c_1 < c_2 \leq n$)) are randomly selected in $M_{j,better}(t) = \left\{X_1^j(t), X_2^j(t), \cdots, X_n^j(t)\right\}$. A portion (including the crossover nodes) between the two crossover nodes is used as a crossover region $C_j(t)$ according to Equation (12). $C_j(t)$ is inserted into the corresponding position of $M_j(t)$. For example, when $C_j(t) = 231$, $M_j(t) = 133524121$, $c_1 = 4$, and $c_2 = 6$ are given, the cross result is $M_{j,c}(t) = 133231121$. The method of crossover is shown in Equation (13). The individual $M_{j,c}(t)$ after the operation is obtained. $M_{j,c}(t)$ is substituted into Equation (10) to obtain the fitness $f_{j,c}(t)$.

$$C_j(t) = \left(X_{c_1}^j(t), X_{c_1+1}^j(t), \cdots, X_{c_2}^j(t)\right), \tag{12}$$

$$\begin{cases} M_j(t) = 133524121, \\ C_j(t) = 231, \end{cases} \rightarrow M_{j,c}(t) = 133231121. \tag{13}$$

### 3.4. Detection Behavior

In the detection behavior of the original beetle antennae search algorithm, the set step size is linearly decreasing with time. The reduction length is fixed, and the adjustment cannot be made according to the actual situation. The iteration speed is slow, making the algorithm easily fall into the local optimum. When the fitness of the right antennae is equal to the left, the beetle does not move, and the algorithm is in an undeveloped state. Therefore, we adjust the original linear decrement parameter of the step size of the beetle to the adaptive dynamic parameter. It enhances the ability of the algorithm to jump out of the local optimum.

Comparing the individual $M_{j,c}(t)$ obtained by the search behavior with the optimal individual $M_{best}(t)$ of the $t$-th generation population, we select the individual with high fitness to update $M_{best}(t)$. The process $M_{best}(t)$ is updated, as shown in Equation (14). The optimal fitness of the current population $f_{max}(t)$ is calculated by Equation (10).

$$M_{best}(t) = \begin{cases} M, & \text{if } f(M_{j,c}(t)) > f(M_{best}(t)), \\ M_{best}(t), & \text{else,} \end{cases} \tag{14}$$

where $M_{best}(t) = M_1(1)$ when $t = 1, j = 1$.

We calculate the overall fitness of the current population, which is sum of all individual fitness $f_{all}(t)$ by Equation (15) and the average fitness $f_{av}(t)$ by Equation (16).

$$f_{all}(t) = \sum_{j=1}^m f_j(t), \tag{15}$$

$$f_{av}(t) = \frac{\sum_{j=1}^m f_j(t)}{m}. \tag{16}$$

BAS-MPE adaptively adjusts the algorithm's direction and the rate of optimization of the algorithm according to the current fitness level $\frac{f_{av}}{f_{max}}$ of the population. When the current fitness of BAS-MPE is at a higher level ($\frac{f_{av}}{f_{max}}$ is higher), individuals need local development in a small area, and the corresponding moving step size should be reduced. The operation of assigning status to all nodes (its number is $n$) is to select the number of changed nodes. The moving step length increases when it is at a lower level. It makes the position change that a large number of beetles in the population move and jump out of the current local optimum. We set an adaptive operator to adjust the step size to the current fitness $\frac{f_{av}}{f_{max}}$. We construct an adaptive step size as shown in Equation (17).

$$\delta = randn\left( n \times (1 - \frac{f_{av}}{f_{max}})^{\lambda} \right), \tag{17}$$

where $\lambda \in (0,1)$ is the dynamic adaptive adjustment factor.

Individuals perform detection behavior based on adaptive moving steps. Select $\delta$ mutation positions $(1, 2, \cdots, \delta)$ and mutate at these positions. The variation range of each node $\left( X_1^j(t), X_2^j(t), \cdots, X_{\delta}^j(t) \right)$ is within the node state range $\left( U_1^j(t), U_2^j(t), \cdots, U_{\delta}^j(t) \right)$ (excluding the original state). The mutation is not performed if the node at the mutation position is the evidence node.

Assign any other node state $u_{w,ra} \in U_w^j(t), (w = 1, 2, \cdots, \delta)$ at the mutation location $X_w^j(t)$ to the mutation node. Combine all nodes' recombination status schemes into one individual $M_j(t)$. The detection behavior produces a new individual process, as shown in Equation (18).

$$\begin{cases} X_i^j(t) = \begin{cases} X_w^j(t) = u_{w,ra}, u_{w,ra} \in U_w^j(t), \text{ if } i{=}{=}w \cap X_i^j(t) \notin \{e_1, e_2, \cdots, e_v\}, \\ X_i^j(t), else, \end{cases} \\ M_j(t) = \left\{ X_i^j(t) \right\}. \end{cases} \tag{18}$$

We calculate the fitness of $M_j(t)$ according to Equation (10) and compare it with $f_{j,c}(t)$, selecting the individual with high fitness as the $j$-th next-generation individual $M_j(t+1)$ generated by the algorithm.

The search behavior and detection behavior generate all individuals in the $t$-th generation of the population $G(t)$, forming a new population, the next generation population $G(t+1)$. When the maximum number of iterations is reached, the algorithm outputs the current optimal individual, MPE.

### 3.5. The Steps and Flowchart of BAS-MPE

#### 3.5.1. The Steps

- Step 1: Input the structure of Bayesian networks (DAG) and conditional probability table (CPT), add evidence (E), and initialize $t = 1$, $t_{max}$.
- Step 2: Generate a discrete chaotic sequence $Z^m = \left( z_1^m, z_2^m, \cdots, z_n^m \right)$. Construct an initial population $G(1)$ by calculating the current chaotic individual.
- Step 3: Calculate the left and right antennae coordinates $M_{j,lt}(t)$ and $M_{j,rt}(t)$ by Equation (9).
- Step 4: Calculate the fitness of $M_{j,lt}(t)$ and $M_{j,rt}(t)$ by Equation (10). Update the better coordinate $M_{j,better}(t)$ by Equation (11) and the fitness $f_{j,c}(t)$.
- Step 5: Calculate the crossover individual $M_{j,c}(t)$ by Equations (12) and (13) and the fitness $f_{j,c}(t)$.
- Step 6: Update the optimal individual $M_{best}(t)$ by Equation (16) and the fitness $f_{max}(t)$.
- Step 7: Calculate the overall fitness level of the current population $f_{all}(t)$ and the average $f_{av}(t)$.

- Step 8: Construct an adaptive step size $\delta$ by Equation (17), update the individual $M_j(t)$ by Equation (18), and the fitness $f_j(t)$ of $M_j(t)$.
- Step 9: Compare the fitness of $M_j(t)$ and $M_{j,c}(t)$, and select individuals with high fitness as the next generation of individual $M_j(t+1)$. Generate new individuals from all individuals $G(t)$ in the $t$-th generation population and form new populations as the next generation population $G(t+1)$.
- Step 10: If the algorithm satisfies $t < t_{max}$, it goes $t = t+1$ and jumps to step 3; otherwise, it outputs the global optimal individual, that is, MPE.

### 3.5.2. The Flowchart

We developed an approach for the most probable explanation in Bayesian networks based on the beetle antennae search algorithm (BAS). In our approach, an individual is associated with the state of all nodes in the network. The node corresponding to each individual uses the search behavior of BAS and learns the left and right antennae in separate directions. Our algorithm sets an adaptive dynamic parameter according to the idea of adaptive factor to adjust the step size of the current individual movement in the detection behavior. This representation provides an advantage since the search range and optimization rate of the algorithm are adaptively adjusted according to the fitness level of the current population. The corresponding selection changes the number of nodes, Ih changes the position and makes many individuals in the population move and jump out of the current local optimum. The flowchart for our approach is shown in Figure 2.

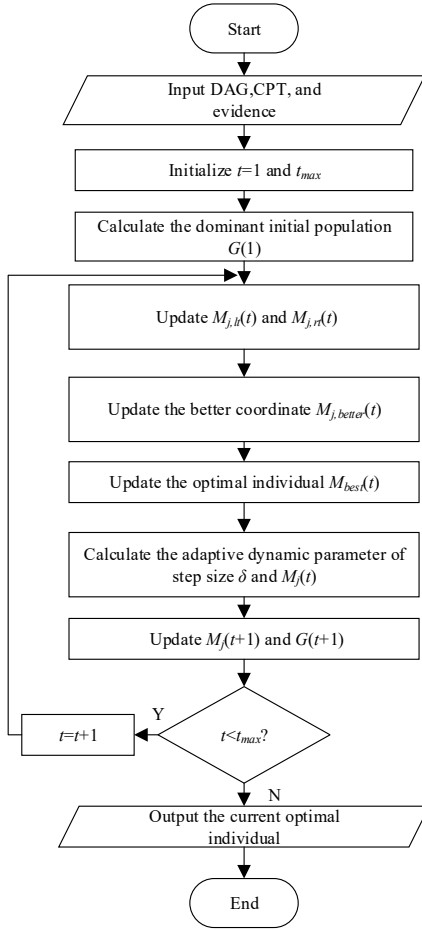

**Figure 2.** The flowchart of BAS-MPE.

The main loop of this algorithm consists of three phasIs. In the first phase (processes 1–3), all individuals in the population are initialized, and the dominant initial population is

generated by the chaos strategy. In the second phase (processes 4–6), all individuals in the population execute a search strategy to generate the left and right antennae coordinates and generate optimal individuals. In the last phase (processes 7–8), our algorithm improves the detection behavior by adjusting the step size based on dynamic parameters, generates the individual, and forms a new population. Finally, our algorithm outputs the optimal individual of the current population when the iterative conditions are finally satisfied through iteration, which is the most probable explanation.

### 3.6. Application of BAS-MPE for the Spindle Process Parameter Optimization of CNC Machine Tools

This section uses the proposed algorithm BAS-MPE based on the beetle antennae search optimization strategy to establish an optimization model for the spindle process parameters of CNC machine tools and conducts experiments in Section 5.3.

The research applied in this section is to maximize the final target machining quality *Quality* by optimizing the three cutting process parameters of CNC: cutting speed $V_c$, feed speed $f_z$, and cutting depth $ap$. In the Bayesian network, we can use the maximization of conditional probability to express this optimization goal. It can be expressed as Equation (19).

$$(V_c, f_z, ap) = \mathrm{argmax}[P(Quality|V_c, f_z, ap)] \tag{19}$$

First, *Quality* is analyzed to clarify the optimization objective function. In the Bayesian network, it is difficult to directly establish the functional expression relationship between $V_c$, $f_z$, $ap$, and *Quality* through equations. Instead, obtaining the probability model diagram outputs the status of the corresponding process parameters under maximum probability. In this study, the maximum probability refers to the maximum conditional probability $P(Quality|V_c, f_z, ap)$. $P(V_c, f_z, ap)$ can be obtained from the data, so it is a fixed value. Therefore, maximizing $P(Quality|V_c, f_z, ap)$ is equivalent to maximizing $P(Quality, V_c, f_z, ap)$, and $P(Quality, V_c, f_z, ap)$ is the objective function of the most possible explanation of Bayesian networks. Therefore, we derive the quality-process parameter optimization objective function in Equation (20).

$$(V_c, f_z, ap) = \mathrm{argmax}[P(Quality, V_c, f_z, ap)] \tag{20}$$

Then, we use the Bayesian network structure and its conditional parameter table to obtain the most probable explanation for modeling. The quality-process parameter Bayesian network structure $DAG$ and its conditional probability table $CPT$ are obtained through the quality-process parameter data $DATA$ and the Bayesian network structure algorithm [23]. This part can be expressed by Equation (21).

$$\langle DAG, CPT \rangle = f(DATA) \tag{21}$$

where $f(*)$ is to use the Bayesian network structure algorithm on $*$.

Finally, based on the quality-process parameter optimization objective function (Equation (20)) and, $DAG$ and $CPT$, the BAS-MPE is used to obtain the process parameter optimization values. The process parameter optimization model is expressed as Equation (22).

$$(V_c, f_z, ap) = \mathrm{BAS\text{-}MPE}_{\max[P(Quality, V_c, f_z, ap)]}(DAG, CPT, t_{max}) \tag{22}$$

where $t_{max}$ is the maximum number of iterations.

The process of establishing the process parameter optimization model is as follows:

- Step 1: Establish the quality-process parameter optimization objective function with Equation (20).
- Step 2: Set the maximum number of iterations.

- Step 3: Obtain the Bayesian network structure and conditional probability table with the Bayesian network structure algorithm and data from the machining spindle of CNC machine tools.
- Step 4: Use the dominant initial population strategy to obtain the solution set of the initial state of the process parameters.
- Step 5: Use the improved search behavior and detection behavior to update and iterate the solution set of the state of the process parameters.
- Step 6: Find the optimal solution with the above steps, which is the obtained range of process parameters.

## 4. Convergence Analysis of BAS-MPE

For the convergence analysis of the algorithm, we use the theories of the Lebesgue measure method and the convergence criteria for random search algorithms to prove that it converges in a region. Firstly, our fitness function is the joint probability. In the iterative process, the algorithm always chooses the individual with a large fitness function as the next-generation individual. Therefore, the fitness value is always monotonous and gradually converges to the supremum of the solution space for the subset of the optimal solution set that satisfies the Borel criterion. Finally, it shows that the probability of our algorithm not finding the individual in the subset in infinite consecutive times is 0. That is, the algorithm must be able to find the subset individual in the finite iterative search. So, it meets the convergence criterion. It shows that there must be an individual in the optimal region after finite iterations.

In this paper, the convergence of BAS-MPE is analyzed by the Lebesgue measure method and the convergence criteria for random search algorithms [35]. When the algorithm satisfies Hypotheses 1 and 2, it trends the optimal solution set in solution space.

**Hypothesis 1.** *If $f(H(z, \xi)) \geq f(z)$ and $\xi \in W$ are established, $f(H(z, \xi)) \geq f(\xi)$ is right in the BAS-MPE.*

where $f$ is the fitness of BAS-MPE. $W$ is the optimal solution set. $\xi$ is a random individual in $W$. $z$ is the upper point of $W$ that can generate the upper bound of the value of the objective function.

**Hypothesis 2.** *In the BAS-MPE, the solution sequence of the algorithm is $\{x_k\}_{k=0}^{+\infty}$. When the Lebesgue measure $\{x_k\}_{k=0}^{+\infty}$ is always greater than 0, $\{x_k\}_{k=0}^{+\infty}$ converges to the optimal solution set $W$ with the probability 1.*

where $W_1$ is a subset of [35].

**Lemma 1.** *BAS-MPE satisfies Hypothesis 1.*

**Proof of Lemma 1.** It is proved that the iteration function $H(*)$ of the algorithm is defined as:

$$H\big(M_{best}(t), M_j(t)\big) = \begin{cases} M_{best}(t), f(M_{best}(t)) \geq f(M_j(t)), \\ g(M_j(t)), f(M_{best}(t)) < f(M_j(t)), \end{cases} \tag{23}$$

where $g(*)$ is the search and detection behavior function of the individual. $g(M_j(t))$ is the position of the beetle individual after the $t$-th updating. $M_{best}(t)$ is the current optimal individual position. □

From Equation (23), the iteration function selects the individual with high fitness as the updated individual and the corresponding fitness value in the algorithm is always monotonous and does not decrease. It gradually converges to the upper solution space. So $f(H(z, \xi)) \geq f(\xi)$ is right.

**Lemma 2.** *BAS-MPE algorithm satisfies Hypothesis 2.*

**Proof of Lemma 2.** For a subset of Borel $W_1$ that belongs to $W$, the probability measure is $v(W_1) > 0$, so Equation (24) holds [36]. □

$$\prod_{n=1}^{\infty}[1 - \mu_n(W_1)] = 0. \tag{24}$$

It shows that the probability of not searching for an individual in an infinite number of consecutive searches is 0. The algorithm must find the subset of individuals in the finite iteration search. According to the F.SOLIS's convergence criterion [37], there is $\lim_{k \to \infty} P[x_k \in W] = 1$. This explains that the probability of the *k*-th step belonging to $x_k$ in the algorithm is 1. It shows that after a finite number of iterations in the algorithm, an individual must be in an individual in the optimal solution set $W$.

According to the proofs of Lemma 1, Lemma 2, and the iterative principle of BAS-MPE, the individuals are close to each other, and the optimal individual converges in the optimal solution set in all subsequent iterations, which satisfies the convergence criterion [35]. The BAS-MPE algorithm converges to the optimal solution set.

Figure 3 shows the iteration curve of the fitness of the optimal individual under the four networks. The algorithm can converge to the global optimal in the first three networks. In the fourth network (the Asia network, Insurance network, Alarm network, and Hailfinder network [23,31,38–40]), there is no convergence to the global optimum in the first 100 generations because of the limitation of the number of iterations. However, the number of increasing iterations can further increase the convergence value. This is because the Hailfinder network is more complex than other networks, and the increase in nodes and edges leads to an exponential increase in the amount of calculation. Therefore, the first 100 generations in our experiment did not reach the convergence state of the network.

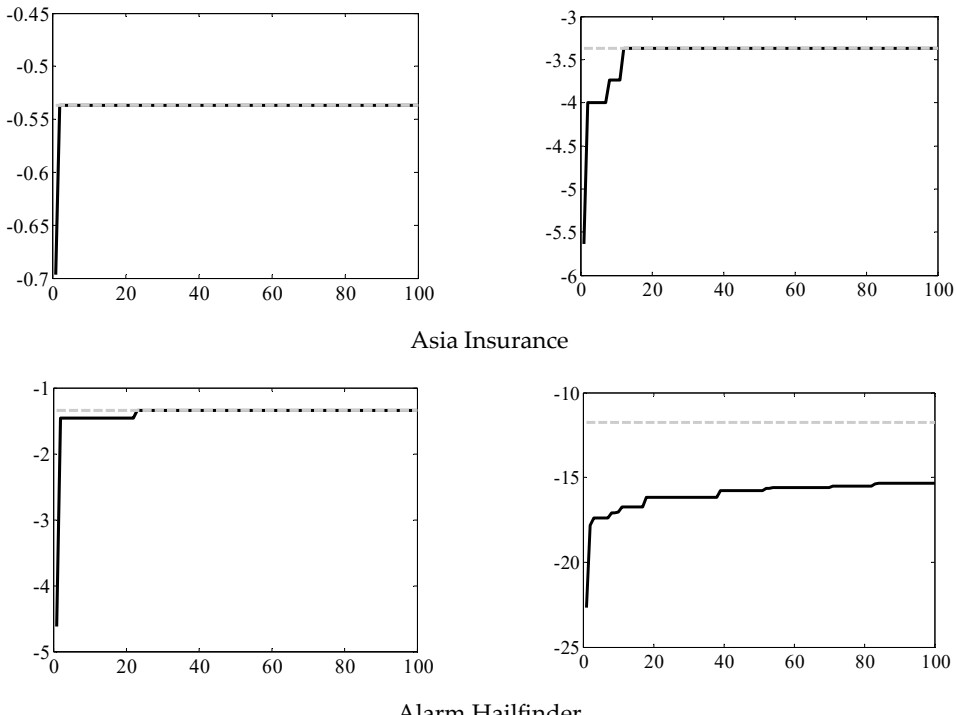

Asia Insurance

Alarm Hailfinder

**Figure 3.** Iterative curve of fitness $f$. The dash line shows the best fitness and the black line shows the optimal individual fitness in BAS-MPE.

## 5. Experiments

The algorithm is implemented in MATLAB. The processor is an Intel (R) Core (TM) i5-3470 CPU, 3.2 GHz, running on a Windows 7 64-bit operating system with 12 GB RAM.

The parameters of this algorithm are set, the maximum number of iterations is 100, and the individual size is 100.

The algorithms are compared in the Asia network, Insurance network, Alarm network, and Hailfinder network. The four "standard networks" used in our experiment are classic network examples commonly used in Bayesian network research. They are also used by many scholars as experimental benchmarks. They can well reflect the results and performance of comparative experiments, such as indicators in time, joint probability, and BIC score. Therefore, they are good benchmarks. These four networks are selected as experimental data set benchmarks in this paper. The experimental results of each algorithm are run 100 times independently to obtain the average value as the final statistical result. This experiment uses a common metric: logarithmic joint probability f [29,34], the formula of which is shown in Equation (10). The larger the value of f, the better the result of the algorithm. The relevant parameters of the classic network are shown in Table 2.

**Table 2.** Classic network related parameters.

| Network | Nodes Size | Edges Size | Nodes Status Range | Maximum Size Parent Nodes |
|---------|-----------|-----------|--------------------|---------------------------|
| Asia | 8 | 8 | 1–2 | 2 |
| Insurance | 27 | 52 | 2–4 | 3 |
| Alarm | 37 | 46 | 2–5 | 4 |
| Hailfinder | 56 | 66 | 2–11 | 4 |

In Table 2, "Nodes size" represents the number of nodes owned by the network, which is essentially the modeling of variables. "Edges size" represents the number of edges owned by the network. Edges are the connection between two nodes and represent the ownership relationship between two nodes. "Nodes status range" represents the discrete state range of each node after discretization. For example, 1–2 means that there are two states, 1 and 2, and 2–5 means that there are four states in total, 2, 3, 4, and 5. "Maximum size parent node" refers to the maximum number of cases in which a node has an edge over another node, and the other node points to the direction of the node among all nodes in the network.

### 5.1. Experimental Comparison between Random and Dominant Initial Population

In this section, we analyze the influence of the dominant initial population on the algorithm using the BAS-MPE algorithm of the dominant initial population and the random initial population. In the case of (without) evidence, the experimental comparison is of the logarithmic joint probability f for MPE in the Asia network, the Insurance network, the Alarm network, and the Hailfinder network. The larger the value of f, the better the result of the algorithm. The results are shown in Table 3.

**Table 3.** Comparison results of logarithmic joint probability under different initial population settings.

| Network | Rand | Advantage | Rand (E) | Advantage (E) |
|---------|------|-----------|----------|---------------|
| Asia | −0.5371 | −0.5371 | −0.2335 | −0.2030 |
| Insurance | −5.7752 | −3.3708 | −6.2039 | −3.9432 |
| Alarm | −4.8818 | −1.3405 | −6.3621 | −1.3405 |
| Hailfinder | −17.2247 | −15.3525 | −18.3291 | −14.5678 |

In Table 3, Rand indicates that the algorithm uses a random initial population without evidence. Advantage demonstrates that the algorithm uses the dominant initial population without evidence. Rand (E) indicates that the algorithm uses a random initial population under two random pieces of evidence. Advantage (E) indicates that the algorithm uses the dominant initial population under two random pieces of evidence.

Table 3 shows that f from the BAS-MPE algorithm with Advantage or Advantage (E) is larger than other algorithms with Rand or Rand (E), so it indicates that the results of the BAS-MPE algorithm using the dominant initial population are better than those using the random initial population. The BAS-MPE algorithm uses a chaotic strategy to increase the population diversity of the previous algorithm and reduce the possibility of the algorithm falling into the local optimum.

*5.2. Experimental Comparison with Other Algorithms*

This section uses the BAS-MPE algorithm, DOSI algorithm [23], ANT-MPE algorithm [29], NGA algorithm [30], SLS algorithm [31], and DMVPSO algorithm [32] to compare the average logarithmic joint probability f-av, maximum logarithmic joint probability f-max, minimum logarithmic joint probability f-min, and running time t(s). Our test data sets are derived from the Asia, Insurance, Alarm, and Hailfinder networks [23,31,38–40]. The results of the experiment are shown in Tables 4–7. We have made the contents of Tables 4–7 into Figure 4 to more intuitively reflect the results of the algorithm comparison.

**Table 4.** Logarithmic joint probability and running time of each algorithm in the Asia network.

| Algorithm | f-Av | f-Max | f-Min | t (s) |
|---|---|---|---|---|
| BAS-MPE | −0.5371 | −0.5371 | −0.5371 | 0.86 |
| ANT-MPE | −0.5371 | −0.5371 | −0.5371 | 1.70 |
| DOSI | −0.5565 | −0.5371 | −0.5693 | 0.93 |
| NGA | −0.5371 | −0.5371 | −0.5371 | 1.57 |
| DMVPSO | −0.5829 | −0.5371 | −0.6127 | 1.25 |
| SLS | −0.5371 | −0.5371 | −0.5371 | 1.21 |

**Table 5.** Logarithmic joint probability and running time of each algorithm in the Insurance network.

| Algorithm | f-Av | f-Max | f-Min | t (s) |
|---|---|---|---|---|
| BAS-MPE | −3.9205 | −3.3708 | −4.0628 | 11.87 |
| ANT-MPE | −7.3592 | −6.9830 | −7.9629 | 21.26 |
| DOSI | −9.2839 | −8.7931 | −11.2709 | 16.76 |
| NGA | −4.2210 | −4.1052 | −4.4928 | 35.77 |
| DMVPSO | −8.5577 | −6.9991 | −12.2983 | 18.93 |
| SLS | −4.5824 | −3.6979 | −5.8406 | 37.98 |

**Table 6.** Logarithmic joint probability and running time of each algorithm in the Alarm network.

| Algorithm | f-Av | f-Max | f-Min | t (s) |
|---|---|---|---|---|
| BAS-MPE | −1.6908 | −1.3405 | −1.8846 | 19.01 |
| ANT-MPE | −2.9151 | −1.9913 | −3.5490 | 39.93 |
| DOSI | −1.7079 | −1.5964 | −2.0034 | 27.38 |
| NGA | −2.5826 | −1.8929 | −2.6637 | 42.47 |
| DMVPSO | −1.9500 | −1.7317 | −2.5379 | 35.57 |
| SLS | −1.9713 | −1.8711 | −2.8853 | 33.42 |

**Table 7.** Logarithmic joint probability and running time of each algorithm in the Hailfinder network.

| Algorithm | f-Av | f-Max | f-Min | t (s) |
|---|---|---|---|---|
| BAS-MPE | −20.7535 | −15.3524 | −22.7938 | 37.29 |
| ANT-MPE | −30.6043 | −24.0157 | −33.6972 | 66.11 |
| DOSI | −50.2858 | −46.8836 | −57.9420 | 40.50 |
| NGA | −90.2771 | −64.2572 | −110.9722 | 46.76 |
| DMVPSO | −136.7425 | −110.2583 | −155.8347 | 42.68 |
| SLS | −144.6733 | −129.6434 | −176.7300 | 55.27 |

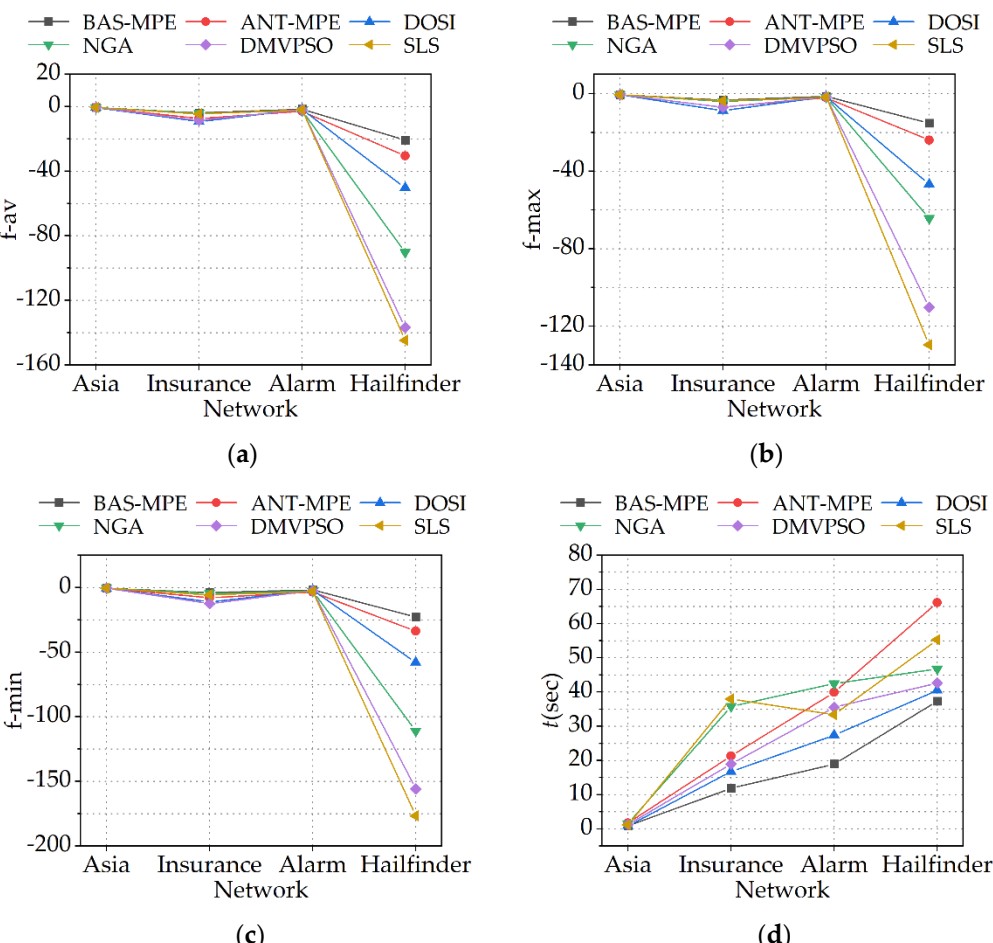

**Figure 4.** Experimental comparison results of the average logarithmic joint probability f-av, maximum logarithmic joint probability f-max, minimum logarithmic joint probability f-min, and running time t. (**a**) The average logarithmic joint probability f-av; (**b**) the maximum logarithmic joint probability f-max; (**c**) the minimum logarithmic joint probability f-min; (**d**) the running time t.

For the DOSI algorithm [23], the number of particles in each swarm was set to 20, and σ was set to 0.2, which is an adjusting parameter. Other settings are consistent with the particle swarm optimization settings in the DMVPSO algorithm [32]. In the ANT-MPE algorithm [29], we used the most appropriate parameter setting from its conclusion. The parameter settings mainly include the weight of pheromone trails, which is 1, the weight of the local heuristic function, which is 5, and the skewness degree, which is 0.1. For the NGA algorithm [30], we still use the parameter settings of the original algorithm. The crossover rate is 1.0 and the mutation rate is 0.06. The SLS algorithm [31] is a variant of a random combination greedy search algorithm, which does not have the characteristics of selection adaptation. So, there is no process of parameter setting. For the DMVPSO algorithm [32], we use the parameter setting of the original algorithm with the learning factors of 2. A discrete control parameter of 0.2 is proposed in this paper. Other parameters, including inertia weight, are obtained adaptively and do not need to be set. The population size is 100, and the number of iterations is 100. In addition to some basic default settings, their parameter settings are the best settings proved by experiments.

Table 4 and Figure 4a–c show that the BAS-MPE algorithm, the ANT-MPE algorithm, the NGA algorithm, and the SLS algorithm can obtain the same maximum logarithmic joint probability of MPE in the Asia network. Their maximum logarithmic joint probability equals the minimum logarithmic joint probability, indicating that several algorithms are more stable in small networks. The average logarithmic joint probability obtained by the DOSI algorithm and the DMVPSO algorithm is poor, and the maximum logarithmic joint

probability and the minimum logarithmic joint probability are not equal, indicating that the stability of the two algorithms is poor.

Table 5 and Figure 4a–c show that in the Insurance network, the average logarithmic joint probability, the maximum logarithmic joint probability, and the minimum logarithmic joint probability of the BAS-MPE algorithm are better than the other five algorithms. This is because the BAS-MPE algorithm improves the search behavior and uses the direction combination sequence as the guidance for generating the required coordinates. The algorithm has multi-directionality in the search process, which makes up for the problem that the algorithm selects the direction individually and avoids the algorithm falling into the local optimum early, so the algorithm finally gets the better results.

Figure 4d shows that the BAS-MPE algorithm runs with less time than the other five algorithms, which indicates that the BAS-MPE algorithm is more efficient in small networks (the Asia network and the Insurance network).

Table 6 and Figure 4a–c show that in the Alarm network, the average logarithmic joint probability of the BAS-MPE algorithm and the DOSI algorithm is close, and both are larger than the other four algorithms. The maximum logarithmic joint probability and the minimum logarithmic joint probability of the DOSI algorithm are smaller than the BAS-MPE algorithm, indicating that the DOSI algorithm has a relatively large range of solution fluctuations. In contrast, the BAS-MPE algorithm obtains better results with a smaller range of solution fluctuations.

Table 7 and Figure 4a–d show that in the Hailfinder network, the results of the BAS-MPE algorithm are far superior to the NGA algorithm, the DMVPSO algorithm, and the SLS algorithm. When the three algorithms are in a large number of network nodes, the search scope of the improved strategy in the previous generation is smaller, and the running time is longer, leading to the algorithm's lower efficiency. The adaptive dynamic parameter of the BAS-MPE algorithm makes the detection behavior of the algorithm more effective, the relationship between the global and the local well balanced, the update speed of the individual population improved, the running time reduced, and the optimization efficiency increased.

Each algorithm calculates the most probable explanation of the top k logarithmic joint probability k-MPE (k = 2, 4, 6, 8) in four standard networks without evidence. We calculate its average logarithmic joint probability f*. Since f* is a negative value, f* takes the absolute value |f*| for comparison (the smaller its value |f*|, the better the algorithm performance), and the experimental results are shown in Figures 5 and 6.

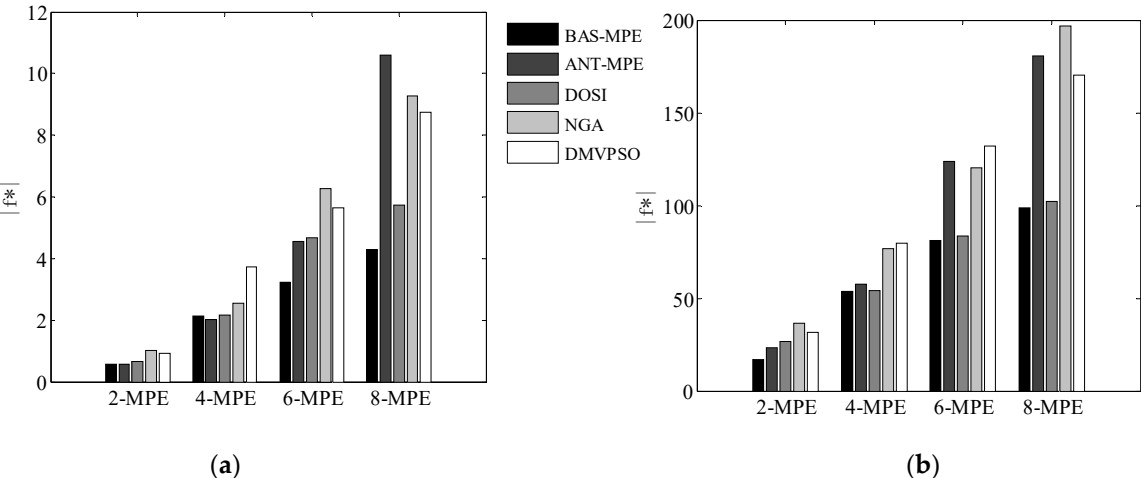

**Figure 5.** Experimental comparison results of |f*| in the Asia network and the Insurance network. (**a**) |f*| in the Asia network; (**b**) |f*| in the Insurance network.

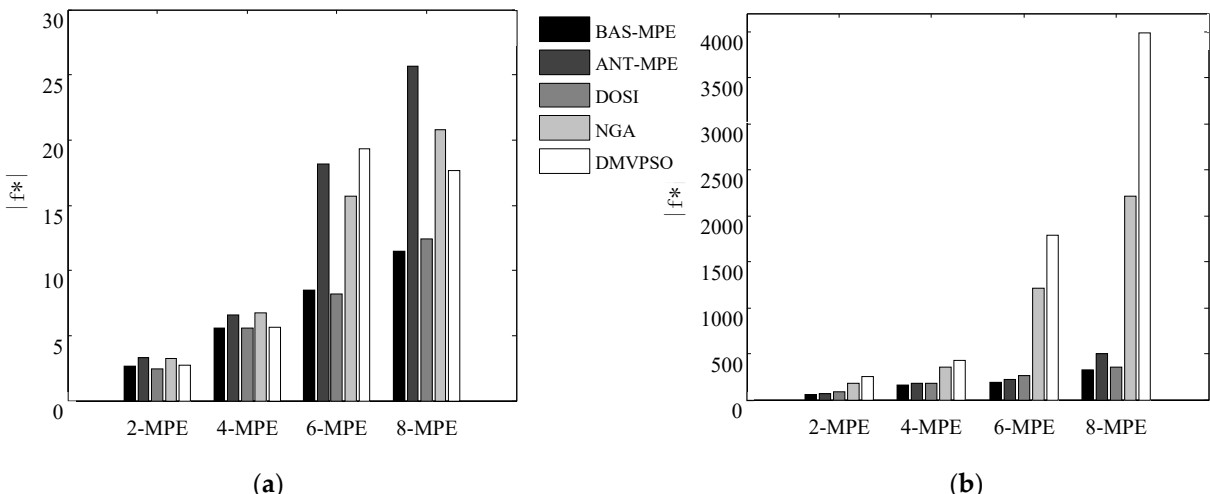

**Figure 6.** Experimental comparison results of |f*| in the Alarm network and the Hailfinder network. (**a**) |f*| in the Alarm network; (**b**) |f*| in the the Hailfinder network.

Figure 5a shows that in the Asia network, the absolute value of the logarithmic joint probability of 2-MPE, 6-MPE, and 8-MPE of the BAS-MPE algorithm is the smallest among the five algorithms. When the 4-MPE is solved, the absolute value of the logarithmic joint probability of the BAS-MPE algorithm is slightly larger than the ANT-MPE algorithm. The ANT-MPE algorithm establishes the ant colony pheromone feedback mechanism and the ant decision table well in the algorithm construction, but the random initial population of the algorithm makes the range of the previous search small, which constantly affects the optimization domain of the algorithm and results in poor overall performance of the algorithm. The BAS-MPE algorithm has the best optimization effect compared to the other four algorithms. The algorithm uses the adaptive dynamic parameter to adjust the step size of the detection behavior. It increases the step size when detecting that the current overall horizontal fitness is at a poor level, which helps the algorithm to jump out of the current local optimum. In the case of ensuring the convergence speed of the algorithm, the ability of global optimization of the algorithm is enhanced.

Figure 5b shows that in the Insurance network, the BAS-MPE algorithm has a lower absolute value of logarithmic joint probability when searching for MPE, which shows that when the overall fitness of the algorithm is the most searching after generation, the level is higher than the other four algorithms. When solving 2-MPE and 4-MPE, the results of the DOSI algorithm are close to the BAS-MPE algorithm. Because the population of the particle swarming optimization framework of the DOI algorithm increases the competition among populations, it has a good preference when selecting individuals, but the algorithm needs more group competition, which affects the efficiency of the algorithm. The improved search behavior and detection behavior of the BAS-MPE algorithm enable all individuals to balance the global search and local development of the algorithm without the need for group competition, which improves the optimization efficiency of the algorithm.

Figure 6a shows that the results of the five algorithms in the Alarm are similar at solving 2-MPE and 4-MPE. As the number of probable explanations to be solved increases, the BAS-MPE and DOSI algorithms show better results. The optimal individuals of the NGA algorithm, the DMVPS algorithm, and the ANT-MPE algorithm are always in rapid evolution when looking for probable explanations. In contrast, other individuals are slow to evolve, and the algorithm cannot get lower. When looking for more probable explanations, the algorithm cannot obtain a lower absolute value of the logarithm joint probability, and the overall fitness of the probable explanations is lower. When looking for 8-MPE, the logarithmic joint probability of the BAS-MPE algorithm has the lowest absolute value, indicating that the algorithm can find the better domain value of a higher-level solution than similar algorithms.

Figure 6b shows that the absolute value of the logarithmic joint probability of the BAS-MPE algorithm in the Hailfinder network is smaller than the other four algorithms. When solving 2-MPE and 4-MPE, the BAS-MPE algorithm, the ANT-MPE algorithm, and the DOSI algorithm can achieve the same logarithmic joint probability absolute value level. But with the number of probable explanations increasing, the BAS-MPE algorithm has a higher fitness level than the ANT-MPE and DOSI algorithms. When solving 6-MPE and 8-MPE, the fitness level of the NGA algorithm and DMVPSO algorithm is rapidly reduced, indicating that these two algorithms are unsuitable for finding more probable explanations of large networks, and the BAS-MPE algorithm can find lower pairs. The search efficiency of the BAS-MPE algorithm is higher than other similar algorithms.

*5.3. An Example of Optimizing Process Parameters of the Spindle Process of Universal CNC Machine Tool CY-K510*

In this section, we use the process parameter optimization model established in Section 3.6 for actual experimental verification.

We collected data on the quality of the tested objects and their corresponding process parameters for the spindle NPH02051 of the universal CNC machine tool CK-Y510. We discretized each quality and process parameter data to them adapt to data modeling. Figure 7 shows the universal CNC machine tool CY-K510 and its spindle NPH02051.

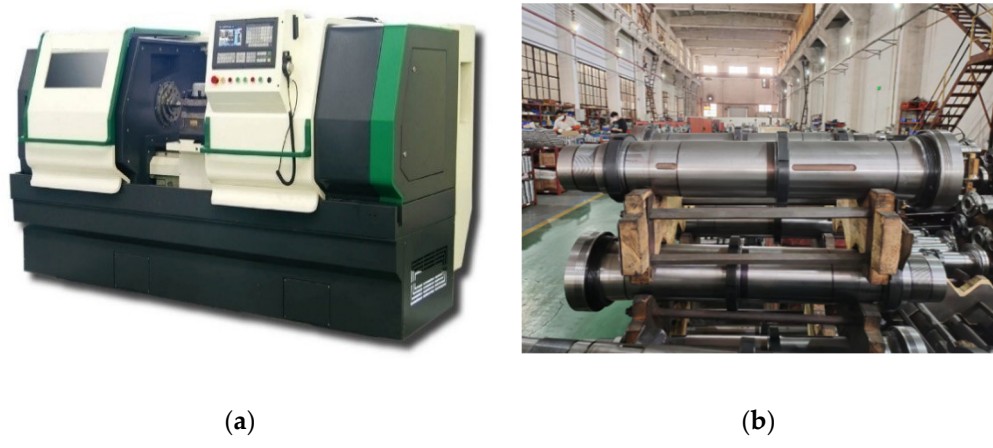

(**a**)　　　　　　　　　　　　　　　　　　　　　　　(**b**)

**Figure 7.** The universal CNC machine tool CY-K510 and its spindle NPH02051. (**a**) The universal CNC machine tool CY-K510; (**b**) the spindle NPH02051 of CY-K510.

Table 8 shows the process parameter optimization model, which includes the following parameters. The cutting tool model, size, material, and structure for specific process steps and tasks in the machining process are basically fixed configurations. Therefore, the adjustment and optimization of cutting speed, feed amount, and cutting depth during the cutting process will affect the final processing quality. The optimization of process parameters in this study includes the calculation of the optimization range of cutting speed $V_c$, feed rate $f_z$, and depth of cutting $ap$. The cutting speed $V_c = \frac{\pi d n}{1000}$ is determined by the rotational speed $n$ and the tool diameter $d$, while the feed rate $f_z = \frac{V_f}{nz}$ is determined by the feed speed $V_f$, the number of gears $z$, and the rotational speed $n$; depth of cutting $ap$ can be directly adjusted.

**Table 8.** The description of process parameters.

| Process Parameters | Composition |
|---|---|
| Cutting tool | Model<br>Size<br>Material<br>Structure |
| Cutting speed | Rotational speed<br>Tool diameter |
| Feed rate | Feed speed<br>The number of gears<br>Rotational speed |
| Cutting depth | / |

The study used data from the spindle NPH02051 of CY-K510 regarding process parameters (which include the three cutting process parameters: cutting speed, feed rate, and depth of cutting), as well as corresponding machining quality data. The optimal range of process parameters is based on the cutting speed, feed rate, and cutting depth corresponding to the maximum joint probability $P$ in the most probable explanation.

Table 9 records the process parameter values of the following seven processing steps before and after optimization using the process parameter optimization model and the corresponding joint probabilities of the most probable explanation. The process parameter values after optimization using the process parameter model are the average values of the output of running 100 experiments. In Table 9, the values (old, new) of the process parameters before and after optimization are expressed as follows: old represents the value of the process parameter before optimization and new represents the value of the process parameter after optimization. It can be seen from Table 9 that the values of the process parameters have changed before and after optimization. Next, we used the process parameter values before and after optimization using the process parameter model in Table 9 as the process input to evaluate the quality results in actual processing.

**Table 9.** Comparison of process parameter values before and after process optimization.

| Work Step Operation | $P$ | $d$ (mm) | $n$ (Old, New) (r/min) | $z$ | $V_f$ (Old, New) (mm/f) | $ap$ (Old, New) (mm) |
|---|---|---|---|---|---|---|
| Cutting cylindrical machining according to size Φ105$h6$ by lathe. | 0.4516 | 105 (Fixed value) | 760, 770 | 1 (Fixed value) | 0.1, 0.12 | 0.6, 0.4 |
| Drill holes according to size Φ85. | 0.2518 | 85 (Fixed value) | 380, 360 | 1 (Fixed value) | 0.09, 0.1 | 0.8, 0.7 |
| Cutting cylindrical machining according to size Φ110$h6$ by lathe and add a margin of 20 mm for processing. | 0.1921 | 110 (Fixed value) | 270, 280 | 1 (Fixed value) | 0.06, 0.08 | 0.9, 0.6 |
| Cutting cylindrical machining according to size Φ110$h6$ by lathe. | 0.2934 | 110 (Fixed value) | 300, 280 | 1 (Fixed value) | 0.08, 0.07 | 0.3, 0.25 |
| Deepen the hole according to dimension Φ82 to half the spindle length. | 0.6425 | 82 (Fixed value) | 610, 630 | 1 (Fixed value) | 0.1, 0.11 | 0.4, 0.3 |
| Expand the range of the hole from Φ23.8$h8$ to Φ23. | 0.2930 | 23.6 (Fixed value) | 670, 680 | 1 (Fixed value) | 0.14, 0.16 | 0.35, 0.2 |
| The workbench mills 16N9 keyway at a 90° working angle. | 0.5546 | 14 (Fixed value) | 2000, 1900 | 1 (Fixed value) | 0.05, 0.04 | 0.4, 0.25 |

We used the process parameters to conduct a machining experiment on the CY company's spindle production line experimental platform. We conducted 100 experimental machining tests, measured the machining quality of the following seven machining work steps, and compared them with the original process parameters. The average quality results were compared, and the comparison results are shown in Table 10.

**Table 10.** The percentage of improvement results for each work step before and after process optimization.

| Work Step Operation | Quality Improvement Results |
| --- | --- |
| Cutting cylindrical machining according to size $\Phi 105h6$ by lathe. | 1.15% |
| Drill holes according to size $\Phi 85$. | 0.95% |
| Cutting cylindrical machining according to size $\Phi 110h6$ by lathe and add a margin of 20 mm for processing. | 0.23% |
| Cutting cylindrical machining according to size $\Phi 110h6$ by lathe. | 2.18% |
| Deepen the hole according to dimension $\Phi 82$ to half the spindle length. | 1.6% |
| Expand the range of the hole from $\Phi 23.8h8$ to $\Phi 23$. | 0.27% |
| The workbench mills 16N9 keyway at a 90° working angle. | 0.59% |

Table 10 shows the optimization results of the process parameters validated on the machining test bench, with improved quality results compared to the measured results. The percentage increase is between 0.23% and 2.18%.

## 6. Conclusions

This paper proposes a most probable explanation algorithm based on an improved beetle antennae search algorithm (BAS-MPE algorithm), which is used to solve the problems of algorithm optimization efficiency being poor and easily falling into local optimum. The BAS-MPE algorithm establishes the dominant initial population by introducing a chaotic strategy, which effectively increases the population diversity. At the same time, the direction sequence is guided as the direction of the individual, avoiding the algorithm falling into the local optimal optimum. The algorithm introduces the adaptive dynamic parameter to adjust the moving step size of the beetle, effectively balances the global and local search relationship, and finds the most probable global explanation by loop iteration. This paper uses the convergence criterion to prove that the BAS-MPE algorithm has good convergence. The results show that the BAS-MPE algorithm can find the most probable explanation of Bayesian networks in the four standard networks by the improved beetle antennae search algorithm. The efficiency and results are better than other similar algorithms and the BAS-MPE algorithm can effectively avoid MPE falling into local optimum. The proposed BAS-MPE algorithm was used to optimize the spindle processing parameters of the universal CNC machine tool CY-K510, and the optimization results were obtained. The optimization results were applied to actual processing experiments, and the quality results were all improved. In the future, the optimization method of CNC machining process parameters studied in this article (optimizing cutting speed, feed speed, and cutting depth to improve machining quality) can be used to improve the ultimate accuracy of industrial mother machines, which is very meaningful. The BAS-MPE used in this article can not only optimize the quality of a single objective but can also be extended to multi-objective optimization and to consider more restrictions. Therefore, the comprehensive consideration of the optimization of process parameters will be further studied.

**Author Contributions:** Data curation, L.Z.; Software, Y.Q.; Formal analysis, L.Z.; Investigation, N.W.; Methodology, E.C.; Resources, L.Z. and N.W.; Writing—original draft, L.Z.; Writing—review and editing, L.Z. and H.L. All authors have read and agreed to the published version of the manuscript.

**Funding:** This research was supported by National Key R&D Program of China (No. 2019YFB1707301).

**Data Availability Statement:** Not applicable.

**Acknowledgments:** The author would like to thank Bin Liu from Yanshan University for his opinions and suggestions in the development process of this work. We would also like to thank Yashuang Huang from Yunnan CY Company for providing the experimental environment and Yun Zhang from Tsinghua University for their suggestions during this research and experimental phase.

**Conflicts of Interest:** The authors declare no conflict of interest.

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
