# Peer review of "An Algorithm for Optimizing the Process Parameters of the Spindle Process of Universal CNC Machine Tools Based on the Most Probable Explanation of Bayesian Networks"

_processes, doi:10.3390/pr11113099_

Round 1

Reviewer 1 Report

Comments and Suggestions for Authors

1. The authors have presented well. It is suggested to add objectives at the end of introduction section.

2.In literature, the authors can brief about process parameters taken for analysis.

Additional Comments: I like reading this article entitled “An Algorithm for Optimizing Process Parameters of the Spindle Process of Universal CNC Machine Tools Based on the Most Probable Explanation of Bayesian Networks”. In this paper an improved algorithm incorporated with the beetle antennae search algorithm has proposed.

Abstract: The abstract requires some additional lines to address specific CNC machine tools parameters that have considered in the manuscript. This paper uses data sets in 4 standard networks to evaluate the time……..(Row no.19th…How the data set is validated from standard values?)

Introduction:

At present, the process of establishing process parameter optimization models adopts multi-objective optimization algorithms …………….(Row no.50th and 51th …Please specify some examples with citations).

The population optimization algorithm is one of the most effective methods to solve MPE based on the local search method.…………..Row 147th and 148th…(Need citation. And why is so?)

Section 3:

Add few applications of the proposed algorithm –BAS-MPE

Section 4:

Row 395th -the convergence of BAS-MPE is analyzed by the Lebesgue measure method and the convergence criteria for random search algorithms……….(Is there any other measure for the analysis?)

Section 6:

The authors should add few lines about the future scope about the research carried out in the paper.

Reviewer 2 Report

Comments and Suggestions for Authors

Strengths of the article are:

- good structure of the presented information;

- visualization of the content: the technological means used are presented;

- there is a significant number of formulas and algorithms;

- there are actually obtained results that are analyzed in detail.

To complete the research it is necessary:

- to update the sources. Only a few were used after 2020, which for such a serious study needs to be adequate to the current trends in the field. This requires adding new sources, possibly comparing good practices and relevance to 2023;

- Figure 7 should be corrected: add a description of the four photos / images with (a), (b), (c) etc. and describe them to the figure; (For example, Figure 5 is correctly represented)

- Similarly (as Figure 7), Figure 4 needs to be improved in terms of description and notations with (a), (b), etc.

- It is desirable to add acknowledgments.

Reviewer 3 Report

Comments and Suggestions for Authors

The manuscript “An Algorithm for Optimizing Process Parameters of the Spindle Process of Universal CNC Machine Tools Based on the Most Probable Explanation of Bayesian Networks” addresses an actual problem related to the optimization of the process parameters to improve the quality of processed parts. An improved algorithm incorporated with the beetle antennae search algorithm for the most probable explanation in Bayesian networks was utilized to achieve optimization calculation of process parameters. Adaptive dynamic step parameters were found to improve detection behavior. The chaotic strategy was discretized and used to establish the dominant initial population in the population initialization. Data sets in four standard networks were employed to evaluate the time and fitness value of the improved beetle antenna search. It was experimentally established that the proposed algorithm is superior in time and accuracy compared to competitive ones. Through optimization of the algorithm, the quality of machining quality was improved.

The manuscript falls within the scope of the journal of Processes.

The title of the manuscript does not correspond to the matter of the study.

The literature survey is very broad (excessive) and should be more focused on the topic of the study.

The algorithms are described with extra details. It is not bad; however, the problem of “Optimizing Process Parameters of the Spindle Process of Universal CNC Machine Tools” were not well stated, solved and described.

Experimental results are very briefly reported.

The problem of “Optimizing Process Parameters of the Spindle Process of Universal CNC Machine Tools” was firstly mentioned in the page 18, just 2 pages before the Conclusion.

The Conclusion in the technical paper must report numerical achievements, rather than a list of conducted studies.

The manuscript requires major revision. The following aspects are to be addressed by the authors.

Page 1, line 35. “attracted people's attention”. What does “people” mean?

Page 2, line 69-76. “The solution of MPE of BNs is used in many fields, and examples are as follows: it queries the state combination of possible explanation of weather conditions through accurate probability inference to obtain the explanation of maximum probability so as to obtain the weather prediction results, it calculates the state combination most likely to explain the disease and accurately realizes the disease diagnosis, the cognitive combination probability of satisfaction relationship is the largest and returns to the optimal combination in the social system, it gives the combination of target detection schemes with the greatest probability to realize fast goal inference based on the target location of pixels [13-17].” It seems that BN’s application for weather forecasting are not very relevant here. The paper is on mechanical engineering problems.

Page 3-4. Section “2. The Related Work on MPE”. The section seems to be excessive. The information might be compressed in moved in the Introduction. In addition, the Introduction might be shortened by removing non-relevant references. In addition, beetle antennae search algorithm (for searching food sources) is described in too many details.

Page 12, line 439 “The Hailfinder network [39], whose nodes are the influencing factors of weather change, is a weather forecast system (synthetic) data set with 56 nodes, 66 edges, and 2656 parameters.”. What is the reason for discussion of the weather forecast system? What is this paper about? In addition, Results and Discussion should not be mixed.

Page 14. Line 495. “the results of the BAS-MPE algorithm using the dominant initial population are better than those using the random initial population”. Are there any numerical estimates?

Page 14, line 50 “5.2. Experimental Comparison with Other Algorithms”. Where the data for algorithms’ testing were taken from?

Page 18, line 628. “5.3. An Example of Optimizing Process Parameters of the Spindle Process of Universal CNC 628 Machine Tool CY-K510”. The manuscript is entitled “An Algorithm for Optimizing Process Parameters of the Spindle Process of Universal CNC Machine Tools Based on the Most Probable Explanation of Bayesian Networks”. However, the first mentioning of CNC machine happens in the end of the text! What was discussed above? Is this paper really about a spindle process?

Page 18. The photographs in figure 7 are mostly senseless. What are they for?

Page 18, Table 8; Page 19 Table 9. It is not very clear, what was optimized. The relevant information is missing.

Page 20, Table 10. How many experiments were performed for each measurement? Where is the information on the data scatter?

Comments on the Quality of English Language

The general level of English language is OK.

Round 2

Reviewer 3 Report

Comments and Suggestions for Authors

The text of the manuscript “An Algorithm for Optimizing Process Parameters of the Spindle Process of Universal CNC Machine Tools Based on the Most Probable Explanation of Bayesian Networks” has been substantially modified. Most of the reviewer’s comments and remarks were taken into account. However, the manuscript still requires a minor revision. The following aspects are to be taken into account by the authors.

 Page 1, lines 11-12 “The three cutting elements (cutting speed, feed speed, and cutting depth) are the most important optimization input parameters”. Those are not cutting elements, but cutting process parameters.

Page 1, lines 13-14. “Therefore, it is necessary to study the three cutting elements of the process parameters of the CNC machine tool spindle”. See the remark above.

Page 3, line 133. “The three cutting elements (cutting speed, feed speed, and cutting depth) are the most critical input parameters”. See the remark above.

Page 11, line 432. “by optimizing the three elements of CNC cutting parameters…”. See the remark above.

Page 17, line 632. “Figure 4. (a) shows the average logarithmic joint probability of each algorithm; (b) shows the maximum logarithmic joint probability of each algorithm; (c) shows the minimum logarithmic joint probability of each algorithm; (d) shows the running time of each algorithm.” Figure caption cannot start from the word “shows”.

Page 17, line 658. “Figure 5. (a) shows the comparison of |f*| in the Asia network; (b) shows the comparison of |f*| in 658 the Insurance network”. Figure caption cannot start from the word “shows”.

Page 18, line 688. “Figure 6. (a) shows the comparison of |f*| in the Alarm network; (b) shows the comparison of |f*| 688 in the the Hailfinder network”. Figure caption cannot start from the word “shows”.

Page 19, line 719. “Figure 7. (a) shows the universal CNC machine tool CY-K510; (b) shows the spindle NPH02051 of 719 the universal CNC machine tool CY-K510”. Figure caption cannot start from the word “shows”.

Page 20, line 733. “include the three cutting elements: cutting speed, feed rate, and depth of cutting”. See the remark above.

Page 20, line 746. “Table 9. Comparison of parameter results before and after process optimization”. What does “parameter results” means?

Comments on the Quality of English Language

The moderate language editing is required.
